# *Salacia chinensis* L. Stem Extract Exerts Antifibrotic Effects on Human Hepatic Stellate Cells through the Inhibition of the TGF-β1-Induced SMAD2/3 Signaling Pathway

**DOI:** 10.3390/ijms20246314

**Published:** 2019-12-13

**Authors:** Mattareeyapar Phaosri, Salinee Jantrapirom, Mingkwan Na Takuathung, Noppamas Soonthornchareonnon, Seewaboon Sireeratawong, Pensiri Buacheen, Pornsiri Pitchakarn, Wutigri Nimlamool, Saranyapin Potikanond

**Affiliations:** 1Department of Pharmacology, Faculty of Medicine, Chiang Mai University, Chiang Mai 50200, Thailand; mattareeyapar@gmail.com (M.P.); salinee.jan@cmu.ac.th (S.J.); mingkwan.n@cmu.ac.th (M.N.T.); seewaboon@gmail.com (S.S.); wutigri.nimlamool@cmu.ac.th (W.N.); 2Graduate School, Chiang Mai University, Chiang Mai 50200, Thailand; pensiri8@hotmail.com; 3Research Center of Pharmaceutical Nanotechnology, Chiang Mai University, Chiang Mai 50200, Thailand; 4Department of Pharmacognosy, Faculty of Pharmacy, Mahidol University, Thung Phaya Thai, Rajathevi, Bangkok 10400, Thailand; noppamas.sup@mahidol.ac.th; 5Department of Biochemistry, Faculty of Medicine, Chiang Mai University, Chiang Mai 50200, Thailand; pornsiri.p@cmu.ac.th

**Keywords:** *Salacia chinensis* L., hepatic fibrosis, hepatic stellate cells, LX-2 cells, transforming growth factor-beta 1

## Abstract

*Salacia chinensis* L. (SC) stems have been used as an ingredient in Thai traditional medicine for treating patients with hepatic fibrosis and liver cirrhosis. However, there is no scientific evidence supporting the antifibrotic effects of SC extract. Therefore, this study aimed to determine the antifibrotic activity of SC stem extract in human hepatic stellate cell-line called LX-2. We found that upon TGF-β1 stimulation, LX-2 cells transformed to a myofibroblast-like phenotype with a noticeable increase in α-SMA and collagen type I production. Interestingly, cells treated with SC extract significantly suppressed α-SMA and collagen type I production and reversed the myofibroblast-like characteristics back to normal. Additionally, TGF-β1 also influenced the development of fibrogenesis by upregulation of MMP-2, TIMP-1, and TIMP-2 and related cellular signaling, such as pSmad2/3, pErk1/2, and pJNK. Surprisingly, SC possesses antifibrotic activity through the suppression of TGF-β1-mediated production of collagen type 1, α-SMA, and the phosphorylation status of Smad2/3, Erk1/2, and JNK. Taken together, the present study provides accumulated information demonstrating the antifibrotic effects of SC stem extract and revealing its potential for development for hepatic fibrosis patients.

## 1. Introduction

The development of hepatic fibrosis is based on an alteration in balanced processes between extracellular matrix (ECM) production and degradation [1]. The primary effector cells that are a key for hepatic fibrogenesis are hepatic stellate cells (HSCs) [2,3]. Normally, HSCs in a quiescent stage produce low level of alpha-smooth muscle actin (α-SMA) and collagen, the markers for fibrosis [4]. In response to liver damage, a variety of paracrine factors, especially transforming growth factor-beta1 (TGF-β1), activate HSC proliferation and transformation into myofibroblast-like cells, which produces excessive amounts of ECM, including collagens (especially types I, III, and V), elastin, glycoproteins, proteoglycans, and hyaluronan [2,5]. The activation of HSCs that increases the ECM remodeling task is a natural process for wound healing in liver tissue [6]. After the injury has subsided, the tissues turn back to the resolution stage, and HSCs become inactive. However, if the damage continues to occur, fibrogenesis is gradually built up and leads to hepatic fibrosis and eventually liver cirrhosis [1]. An increase in ECM accumulation and a decrease in matrix degradation result in the progression of hepatic fibrosis [7]. 

The role of HSCs to degrade ECM is dependent on matrix metalloprotease (MMP) production [8]. The expressions of MMP-2 (known as gelatinase-A) and MMP-9 (known as gelatinase-B) are significantly upregulated in liver fibrosis for ECM remodeling [9]. During HSCs activation and before increased collagen type I expression, HSCs produce the physiological tissue inhibitors of the MMPs (TIMPs), particularly TIMP-1 and TIMP-2 [10]. Particularly, TIMP-1 production is enhanced upon stimulation through TGF-β1 signaling pathway, which is mediated by the activation of TGF-β receptor and the activation of the major downstream molecules (SMAD2/3 phosphorylation) [11,12]. Previous studies have demonstrated that inhibition of the TGF-β1 signaling pathway attenuates liver fibrosis [12,13,14]. In addition, the mitogen activated protein kinases (MAPK) family, including the three major subgroups (extracellular signal-regulated kinase (ERK), p38, and c-Jun N-terminal kinase/stress-activated protein kinase (JNK)), are involved in the proliferation and activation of HSCs and the aggravation of hepatic fibrosis [15]. Interestingly, the prevention of proliferation and migration of HSCs may be key strategies to reduce the progression of hepatic fibrosis [16,17]. However, there is no standard treatment for hepatic fibrosis. Recently, drug discovery for fibrosis treatment is focusing on interfering with TGF-β signaling to reduce hepatic inflammation, inhibit stellate cell activation, and stimulate matrix degradation [6,18].

Alternative medicine has emerged as an interesting means for treating hepatic fibrosis. The water extract of *Salacia chinensis* L. (SC) stem or ‘Kumpang jed chan’ in Thai has been used as a folk remedy to treat patients with cirrhosis in a local hospital with promising results. All parts of this plant contain many biologically active compounds, such as triterpenes, phenolic compounds, flavonoids, glycosides, condensed tannin, steroids, xanthone glucoside, and mangiferin [19,20,21,22], which show diverse medicinal properties, including antioxidant, hypoglycemic, and antiobesity activity [21,23,24]. Although promising results of SC stem water extract have been demonstrated in hepatic fibrotic patients, there is no scientific evidence revealing the effects of SC stem water extract on hepatic fibrosis thus far. 

Therefore, this study aimed to determine antifibrotic activities of SC stem extract and its possible mechanisms of action. The human HSC cell line, LX-2, was used to explore the antifibrotic effects of SC stem extract upon TGF-β1 activation by observing several markers, including α-SMA and collagen type I production, the regulation and activity of MMP-9, MMP-2, TIMP-1, and TIMP-2, and multiple signaling transduction pathways, including SMAD2/3 and MAPK.

## 2. Results

### 2.1. Salacia chinensis L. (SC) Stem Extract Reverses Morphology of HSCs Activation and Suppresses Its Migration via TGF-β1 

The extraction of *Salacia chinensis* L. (SC) stem provided yield of SC extract at 7.35% w/w. To establish the HPLC fingerprint chromatogram for quality control of the SC stem extract, five phenolic acids were quantitatively analyzed. The results found that the SC extract contained at least gallic acid (0.38 ± 0.007 mg/g extract), as seen in Appendix A.

HSC activation is one of the critical processes during hepatic fibrosis. LX-2 cells, which are characterized as an HSC cell line, have been used in this study. Firstly, the cytotoxicity of SC extract in the concentration range of 0–0.1 mg/mL was determined by a colorimetric MTT assay. After 48 h incubation, all concentrations of SC stem extract were not toxic to the cells, as seen in Figure 1A. We selected three concentrations of SC stem extract (0.01, 0.05, and 0.1 mg/mL) for further investigation. Our initial investigation explored whether SC stem extract suppresses activated HSC transformation in response to TGF-β1 stimulation. The schematic is shown in Figure 1B. Normally, LX-2 cells without TGF-β1 stimulation (UT) show the characteristics of unactivated primary HSCs, as seen in Figure 1C. In contrast, when TGF-β1 at 2 ng/mL was added to the cells and incubated for 24 h, the cells strongly exhibited the activated form of HSCs, as seen in Figure 1C. Specifically, TGF-β1-stimulated HSCs showed morphological changes recognized as myofibroblasts with remarkable cellular structure reorganization, including clumping, stretching, and generating many obvious cell-free areas, as seen in Figure 1C. Interestingly, after 48 h SC stem extract treatment, we noticed dramatic changes in cell morphology and cell–cell contact reorganization in a concentration-dependent manner, as seen in Figure 1C. These observations were clearly seen when the concentration of SC extract was increased. In particular, cells treated with SC extract at 0.1 mg/mL exhibited the overall characteristics of untreated cells, as seen in Figure 1C. 

HSC migration has been identified as an initial process which causes hepatic tissue remodeling and fibrotic progression [25]. We identified the effect of SC stem extract on HSCs migration by observing its ability to inhibit the wound healing process. The effect of SC suppression of HSC migration was clearly observed 48 h after the treatment in a concentration-dependent manner. The highest SC concentration at 0.1 mg/mL revealed nearly 75% suppression compared to the untreated cells, as seen in Figure 2A,B. 

### 2.2. Salacia chinensis L. (SC) Stem Extract Suppresses Fibrotic Markers at the Gene Expression Level

Based on the observation that SC extract suppresses HSCs migration and can prevent TGF-β1-induced fibrotic-related morphological changes, we hypothesized that the extract may also have the ability to suppress important fibrotic markers at the gene expression level. To test our hypothesis, the mRNA expression level of *α*-*SMA*, *COL1A1*, *MMP*-*2*, *MMP*-*9*, *TIMP*-*1*, and *TGF*-*β1* was determined by qRT-PCR using *GAPDH* as an internal control. The normal HSCs expressed very low levels of all fibrotic-related genes. After incubation for 24 h with 2 ng/mL TGF-β1, the expression of all selected genes was undoubtedly increased, as seen in Figure 3A–F, compared to untreated cells. Interestingly, the extract at all concentrations could suppress the expression of all selected genes, except *MMP*-*9*, as seen in Figure 3A–F. These results suggest that SC stem extract initially suppresses fibrotic markers at the transcriptional level. 

### 2.3. Translational Levels of Fibrotic Markers Are Suppressed by Salacia chinensis L. (SC) Stem Extract

Since we observed the reduction of major fibrotic-related gene expression, including α*-SMA* and *COL1A1* after SC treatment, we further investigated the existence of these two proteins by immunofluorescence study and confirmed the protein levels by western blot analysis. In untreated cells, α-SMA and COL1A1 proteins were produced at low levels and α-SMA, especially, was observed as a diffused signal in the cytoplasm of the cells, as seen in Figure 4A. TGF-β1 prominently induced both α-SMA and collagen production, as shown in Figure 4A. In addition, the increment of α-SMA in TGF-β1 treated cells was likely transformed from diffused forms to fibrillar structures, as seen in Figure 4A. As expected, the individual cells with SC treatment had a decreased intensity of α-SMA and COL1A1 signals, as seen in Figure 4A. Western blot analysis also confirmed that SC stem extract could dramatically reduce TGF-β1-stimulated production of α-SMA as well as COL1A1 levels, as seen in Figure 4B,C, in a concentration-dependent manner. 

### 2.4. Salacia Chinensis L. (SC) Stem Extract Reduced ECM Components 

Since dysregulation of extracellular matrix (ECM) components is an important factor in hepatic fibrotic progression, we investigated the effects of SC stem extract on the balance of major ECMs. In particular, the level of MMP-2 has been reported to be increased in liver fibrosis [26,27]. This report was supported by our finding that the expression of the *MMP-2* gene was tremendously increased in response to TGF-β1 stimulation, as seen in Figure 3C. According to our observation that the extract at all concentrations could effectively reduce MMP-2 expression, as seen in Figure 5A,C, we therefore performed gelatin zymography to verify whether the activity of MMP-2 is also decreased in SC stem extract-treated cells. Undoubtedly, the results showed that the extract could be able to reduce the activity of MMP-2 compared to that of TGF-β1-treated cells, and this observation is seen in a concentration-dependent manner, as seen in Figure 5A,C. Specifically, TGF-β1 increased the activity of MMP-9 compared to that of the untreated cells, whereas the extract at all concentrations additionally increased the TGF-β1-induced MMP-9 activity, suggesting an ability of SC extract to increase MMP-9 activity, as seen in Figure 5A,B. Moreover, we further investigated whether the activity of both MMP-2 and MMP-9 is affected by their protein production level by performing western blot analysis to evaluate the existence of MMP-2 and MMP-9. As expected, MMP-9 protein was reduced in SC extract treated cells, as seen in Figure 5F,G, whereas MMP-2 protein was increased, as seen in Figure 5F,H. The major proteins that regulate the function of MMP-2 and MMP-9 are TIMP-1 and TIMP-2, which are profibrotic factors in the liver, and they have the ability to enhance fibrotic processes without affecting collagen synthesis. We wonder whether TIMP-1 and TIMP-2 levels could be affected by changes in MMP-2 and MMP-9 activity. After TGF-β1 stimulation, we observed the increase of TIMP-1 and TIMP-2 protein levels and the extract significantly reduced both TIMP-1 and TIMP-2 protein levels in a concentration-dependent manner, as seen in Figure 5A,D,E. However, the ratios of MMP-1:TIMP-1 and MMP-9:TIMP-1 in SC-treated cells at 0.1 mg/mL were increased, as seen in Table 1. This result indicated that the majority of MMP-1 and MMP-9 were not inhibited, with consequently enhanced collagen degradation.

### 2.5. Salacia chinensis L. (SC) Stem Extract Ameliorates the Phosphorylations of SMAD2/3, ERK1/2, and p38 Induced by TGF-β1 

TGF-β1 can stimulate the receptor and convey the signal transduction via several different cascades. SMAD2/3 and MAPKs are major responsible signal transduction pathways of TGF-β1. We therefore evaluated whether SC stem extract exerts its antifibrotic ability through the suppression of these two pathways. We found that cells treated with TGF-β1 showed an increase in SMAD2/3, ERK1/2 and p38 phosphorylation compared to those of untreated condition, as seen in Figure 6A–E. Interestingly, SC extract at 0.01, 0.05, and 0.1 mg/mL significantly reduced phosphorylation of SMAD2/3, ERK1/2, and p38, as seen in Figure 6A–E. 

## 3. Discussion

Fibrosis and cirrhosis of the liver have become major causes of morbidity and mortality across the world [27,28]. Currently, there is no standard treatment for liver fibrosis. Liver transplant has shown improvement of both survival and quality of life in patients with cirrhosis, but this treatment is invasive, expensive, and cannot be performed for all fibrotic and cirrhosis patients [5]. 

Fortunately, a wide range of biological activities offered by natural products and herbal medicines has emerged as one of potential treatments for hepatic fibrosis [7]. *Salacia chinensis* L. has been incorporated into Thai folk medicine for the treatment of hepatic cirrhosis since 2007 [8]. Previous studies have demonstrated that lignans, eleutheroside E2, and 7*R*, 8*S*-dihydrodehyrodiconideryl alcohol 4-*O*-β-*D*-glucopyranoside, which are constituents isolated from SC leaves by methanol extraction, exhibit hepatoprotective effects in primary cultured mouse hepatocytes [29]. However, the Thai traditional formulation for hepatic fibrosis treatment contains *Salacia chinensis* L. stem water extraction, in which gallic acid was found to be a main component. Moreover, the recent publication revealed the effects of gallic acid on hepatic fibrosis by increasing HSCs apoptosis and reducing hepatocyte oxidative stress [30]. These results suggest that the antifibrotic activities of *Salacia chinensis* L. stem water extract possibly come from gallic acid. 

TGF-β1 is a member of the TGF-β superfamily, playing a critical role in the development of hepatic fibrosis [31]. TGF-β activates the TGF-β receptor [32] and relays signal transduction through phosphorylation of downstream effectors, such as SMAD2/3 [12,33], and we found that SC stem extract significantly inhibited phosphorylation of SMAD2/3 and reduced mRNA expression of *TGF-*β1. These results suggest that the extract can inhibit the TGF-β1 activation of the SMAD-dependent signal transduction pathway. It has been shown that ERK can also phosphorylate the linker regions of SMAD1 and SMAD2/3, and consequently inhibit ligand-induced nuclear translocation of SMADs, thereby promoting the TGF-β antiproliferative response [34,35]. Similarly, JNK phosphorylates SMAD3 outside its SXS motif in response to mitogenic and stress signals, leading to enhanced activation and nuclear translocation of SMAD3 [36]. In addition, activation of MAPK kinase 1, which is an activator of JNK, ERK, and p38, results in phosphorylation and activation of SMAD2 [37]. Our data showed that even though cells were stimulated with TGF-β1, the SC treatment could potently inhibit phosphorylation of members of the MAPK kinase pathway, including JNK, ERK, and p38. These results are promising for identifying possible mechanisms of action of *Salacia chinensis* L. in suppressing the production of fibrotic markers in response to TGF-β1 activation. 

MMP-2 activity and production were reduced in SC stem extract treated cells, while the activity and production of MMP-9 were increased in a concentration-dependent manner. These observations strongly suggest that the extract has the ability to suppress MMP-2, but induce MMP-9 expression and function. It is well known that at the early stage of hepatic fibrosis, the expression of MMP-2 is very high [38]. Therefore, agents proved to be potent inhibitors of MMP-2 may be beneficial for hepatic fibrosis treatment. SC extract has potential to be developed as an antifibrotic agent. 

Besides metalloproteinases, other molecules are involved in ECM accumulation which is a common presentation of liver fibrosis [1,16,17,39,40]. TIMPs are a class of secreted enzymes that play a crucial role in deposition and breakdown of ECM [41,42]. TIMP-1 and TIMP-2 are known to be profibrotic factors in the liver [10,43]. It has been reported that liver-specific overexpression of TIMP-1 leads to more severe fibrosis without a significant effect on collagen synthesis [44]. The relationship of TIMP-1 and MMP-9 can be seen by the fact that TIMP-1 can inhibit MMP-9. Therefore, if TIMP-1 is inhibited, the level of MMP-9 activity is normally increased, and thus apoptosis is induced [9,26]. Consistent with the observation that the extract increased MMP-9 production and activity, it also significantly suppressed TIMP-1. In contrast to TIMP-1, TIMP-2 functions as both an MMP inhibitor and activator depending upon its level of production [45,46]. TIMP-2 can inhibit MMP-2 activity by blocking MMP-2 enzyme catalytic site [45]. However, if TIMP-2 is highly produced (like in the case of patients with liver fibrosis, where the level of serum MMP-2 is also high), it can bind to the C terminal region of newly synthesized MMP-2 being transported to the plasma membrane, where this dimer interacts with the membrane-type MMPs, which activates the activity of MMP-2 [7,45,46,47,48]. Our observation that SC stem extract could suppress TIMP-2 and MMP-2 production strongly suggests that the extract may be able to inhibit hepatic fibrogenesis at least in part through the suppression of MMP-2 activation due to the reduction of TIMP-2 level. 

We are the first group to reveal the effects of SC stem extract on antifibrotic activities by modulating the fibrogenesis processes of HSCs from the gene through protein levels. The extract markedly attenuated TGF-β1-stimulated HSC activation. Crucial molecular factors, including α-SMA, COL1A1, TIMP-1, TIMP-2, MMP-2, and TGF-β1 that normally contribute to fibrosis in HSCs were proved to be significantly reduced by the effect of the extract, whereas the extract tends to increase the production and activity of MMP-9. Our findings provide evidence that *Salacia chinensis* L. stem extract may have potential for development as an antihepatic fibrosis treatment in the future. 

## 4. Materials and Methods 

### 4.1. Preparation and Characterization of SC Stem Extract 

The *S. chinensis* L.’s voucher specimen number, PBM 04927, was authenticated and deposited at the Faculty of Pharmacy, Mahidol University, Thailand. For preparing the extract, 60 g of dried SC stem was pulverized into a coarse powder and boiled with 600 mL of distilled water at 100 °C by reflux twice for 2 h. The filtrate was evaporated and spray-dried at input temperature of 185 °C (temperature of the heater) and then at an output temperature of 86 °C (temperature of the extract). The extract was dark brown and dissolved freely in water.

Determination of phenolic content in the extract was determined by HPLC using a C18 column (250 mm × 4.6 mm, 5 µm). Gradient elution was performed using two solvents; A (1% acetic acid in water) and B (100% acetonitrile) for detection. Twenty microliters of the 10 mg of SC stem extract was dissolved in 1 mL of diH_2_O and injected into the column with a flow rate 0.7 mL/min and detection at 280 nm. Peak area and retention time of the extract sample were determined in the comparison with standard curves of various concentration of standard chlorogenic acid, gallic acid, ferrulic acid, mangiferin, and vanillic acid.

### 4.2. Cell Culture

The human HSC cell line, LX-2, was obtained from Merck (Darmstadt, Germany). Cells were cultured in Dulbecco’s modified Eagle’s medium (DMEM; Gibco, Thermo Fisher Scientific, Waltham, MA, USA), supplemented with 2 mM L-glutamine, 2% fetal bovine serum (FBS; Capricorn Scientific, Ebbsdorfer Grund, Germany), 100 U/mL penicillin, and 100 μg/mL streptomycin (Gibco, Thermo Fisher Scientific, Waltham, MA, USA) and incubated at 37 °C in a 5% CO_2_ humidified incubator. Cells were subcultured at approximately 80% confluent (3–4 days after plating cells) by 0.25% trypsin–EDTA (Gibco, Thermo Fisher Scientific, Waltham, MA, USA). 

### 4.3. Cell Viability Assay

LX-2 cells were plated in a 96-well plate at a density of 5 × 10^4^ cells/well and cultured in DMEM supplemented with 2% FBS for 24 h. After that, SC extract (0–0.1 mg/mL) was added into each well and incubated for 48 h. Cell viability was measured by MTT assay. In short, 20 mg/mL of MTT in PBS was added directly to each well and incubated at 37 °C, 5 %CO_2_ for 1–4 h. Then, the purple formazan was solubilized by DMSO. The color intensity was measured at 570 nm by a microplate reader (M965, Metertech Inc., Taiwan). The experiment was performed in triplicate.

### 4.4. Cell Migration Assay

To determine the cell migration ability of LX-2 cells with and without SC extract. LX-2 cells were seeded and cultured as described in Section 4.3. After the scratch wound was made, cells were treated with SC extracts for 48 h. The migration of cells was determined at different time points (0, 24, and 48 h). The closing of scratched wounds was considered to be the completion of the migration process. The migrated areas were analyzed and determined using ImageJ software.

### 4.5. Immunofluorescence Study

To determine the level of collagen I and α-SMA protein production, immunofluorescence study was performed. Briefly, the cells were stimulated with 2 ng/mL of TGF-β1 for 24 h at 37 °C 5% CO_2_ and then treated with SC extract with the presence of 2 ng/mL of TGF-β1 for 48 h. After treatment, cells were fixed with 4% paraformaldehyde in PBS for 15 min and washed with PBS. Next, cells were permeabilized by 0.3% Triton-X 100 in PBS for 5 min and washed again with PBS. Then, cells were incubated with blocking buffer (1% BSA in PBS) for 1 h and with 1:100 of primary antibodies (collagen-1 and α-SMA) overnight at 4 °C in a moist chamber. An antimouse antibody conjugated with Alexa-488 (Thermo Fisher Scientific, Waltham, MA, USA) with the presence of 5 µg/mL Hoechst 33342 (for nuclei staining) (Thermo Fisher Scientific, Waltham, MA, USA) at 1:400 were used to incubate 1 h at room temperature (RT) in the dark. After washing, cells were mounted and then visualized under a fluorescence microscopy (microscope, AX70 Olympus R, Japan).

### 4.6. Western Blot Analysis

Cells were treated and incubated as described in Section 4.5. After that, cells were lysed with 1× reducing Laemmli buffer, collected and heated at 95 °C for 5 min, then separated by 10% SDS-PAGE and transferred to PVDF membranes. The membranes were blocked with 5% nonfat milk in PBS containing 0.05% tween-20 for 1 h at RT and then incubated with 1:1000 of primary antibodies at 4 °C overnight. Anti-α-SMA antibody was obtained from Sigma-Aldrich (Singapore). Anti p-SMAD2/3, p-p38, pJNK, pERK1/2, p38, JNK, ERK1/2, COL1A1, MMP-2, MMP-9, TIMP-1, and TIMP-2 antibodies were purchased from Cell Signaling Technology (USA). The day after primary antibody incubation, membranes were incubated with 1:5000 secondary antibodies (IRDye^®^ 800cw antimouse or IRDye^®^ 680RD antirabbit IgG). The protein immunoreactive bands were visualized by LI-COR odyssey CLx western blotting detector (LI-COR, Lincoln, NE, USA) and quantitated using Image J software. 

### 4.7. Gelatin Zymography Assay

The activity of MMP was evaluated by gelatin zymography assay. Cells were treated and incubated as described in Section 4.5. Then, the supernatants were collected and electrophoresed on gelatin-containing 10% SDS–polyacrylamide gels in a cold running condition. The gels were washed twice in 2.5% TritonX-100 and incubated for 30 min followed by brief rinsing with 10 mM Tris, pH 8.0. The gels were incubated overnight at 37 °C in gelatinase buffer (50 mM, Tris HCL, and 10 mM CaCl_2_ pH 8). Then, the gels were stained with 0.5% (w/v) Coomassie blue R250 in 50% methanol and 10% glacial acetic acid for 30 min, and destained. The clear zones of gelatin digestion indicate MMP-2 and MMP-9 activity. After scanning the gel, the area of clear zones was analyzed by ImageJ software. 

### 4.8. Quantitative Real-Time PCR

The expression of *α-SMA*, *COL1A1*, *MMP-2*, *MMP-9*, *TIMP-1*, and *TGF-β1* genes was detected by quantitative real-time PCR. Briefly, cells were treated and incubated as described in Section 4.5. Total RNA was extracted using an RNA extraction kit (Qiagen, Hilden, Germany). Real-time PCR reaction was performed using ReverTra Ace^®^ qPCR RT master mix with gDNA Remover protocol. The thermocycler setting was 2 min at 95 °C for polymerase activation, 15 s at 95 °C, 15 s at 58 °C, and 1 min at 72 °C for 40 cycles of amplification on the PCRmax Eco 48 real time PCR system (Staffordshire, ST15 OSA, UK). GAPDH was used as reference mRNA. Samples were analyzed in triplicate for each set of primers. The oligonucleotide PCR primers used in real-time PCR (Thermo Fisher Scientific, Waltham, MA, USA) are shown in Table 2: 

### 4.9. Data and Statistical Analysis 

Data were expressed as mean ± standard deviation. One-way ANOVA followed by least-significant difference (LSD) post-hoc analysis was used to determine the difference between groups. *p* values less than 0.05 were considered statistically significant.

## Figures and Tables

**Figure 1 ijms-20-06314-f001:**
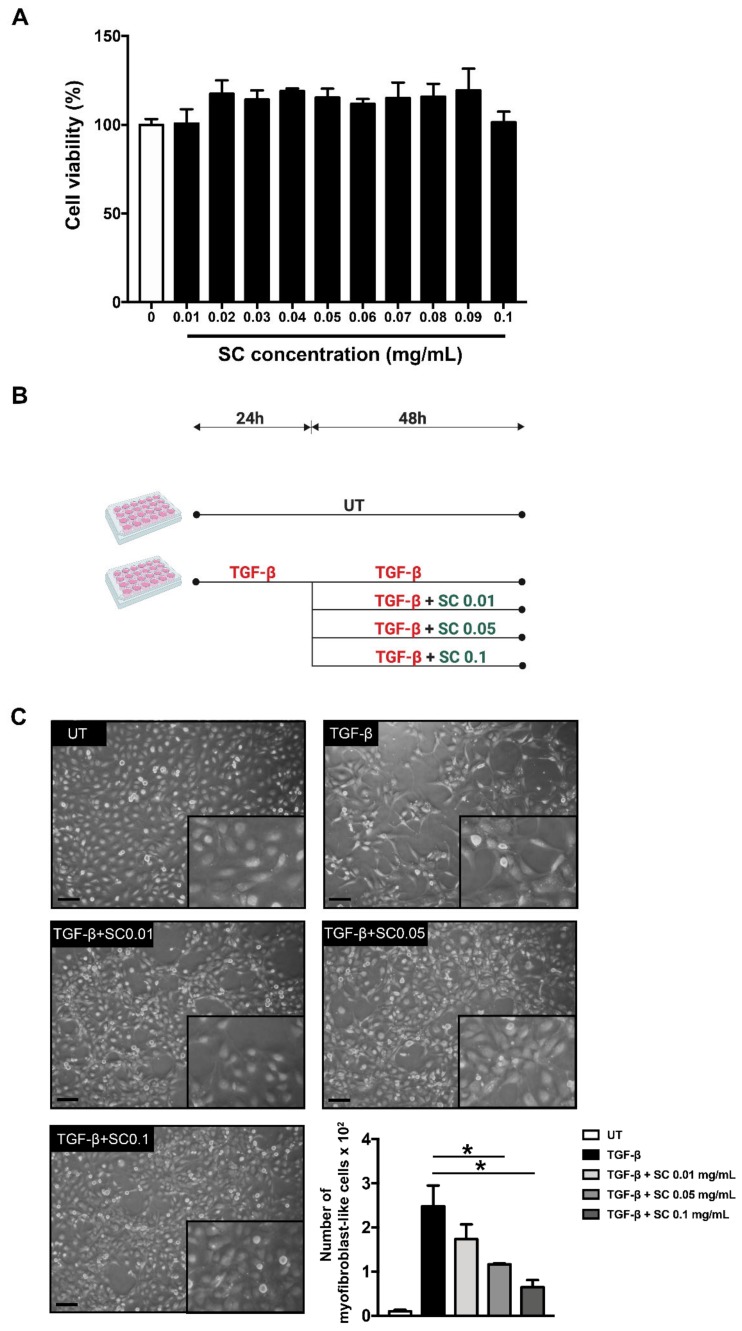
Effects of SC stem extract on stimulated hepatic stellate cell (HSC) morphology. Cell viability of LX-2 cells after SC treatments at 0.01 to 0.1 mg/mL are shown in (**A**). A schematic diagram of the experimental design is shown in (**B**). LX-2 morphology observed under a phase contrast microscopy is shown in (**C**). UT: untreated; TGF-β: TGF-β1 2 ng/mL; SC: *Salacia chinensis* L. extract. Scale bar = 50 µm. * *p* < 0.05.

**Figure 2 ijms-20-06314-f002:**
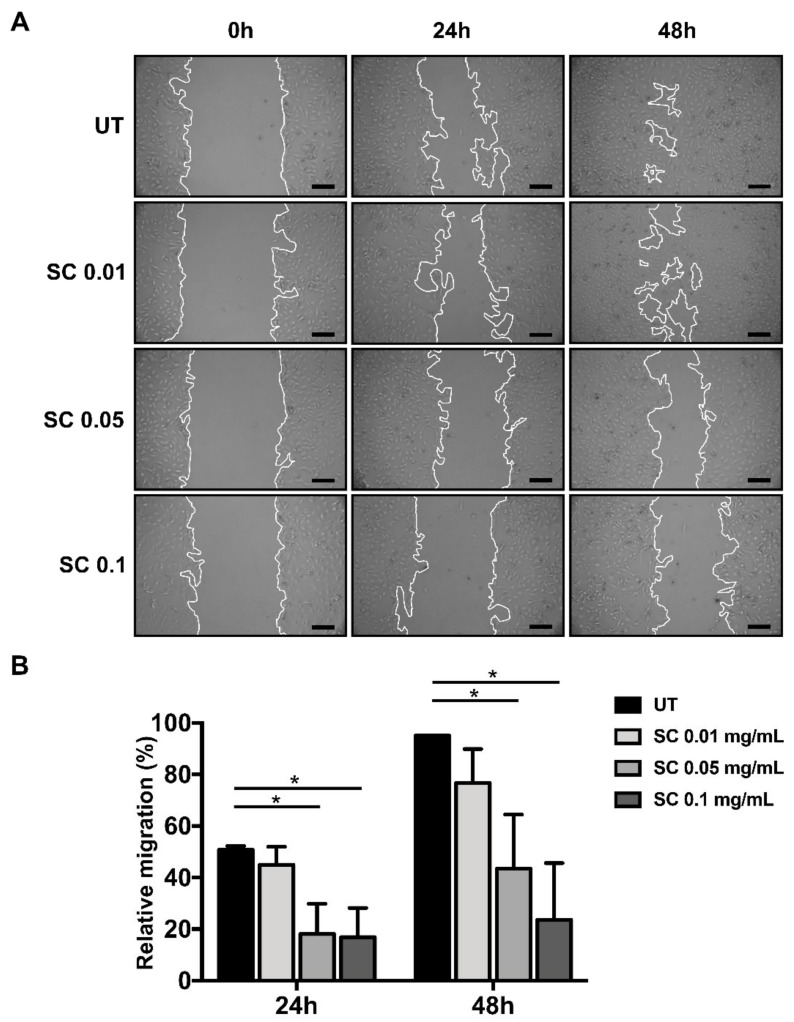
Effects of SC stem extract on HSC 2D migration. Cell migration was observed at 0 h, 24 h, and 48 h after SC treatments (**A**). The percentage of cell migration at 24 h and 48 h after SC treatments is quantified in (**B**). UT: untreated; SC: *Salacia chinensis* L. extract. Scale bar = 50 µm. * *p* < 0.05.

**Figure 3 ijms-20-06314-f003:**
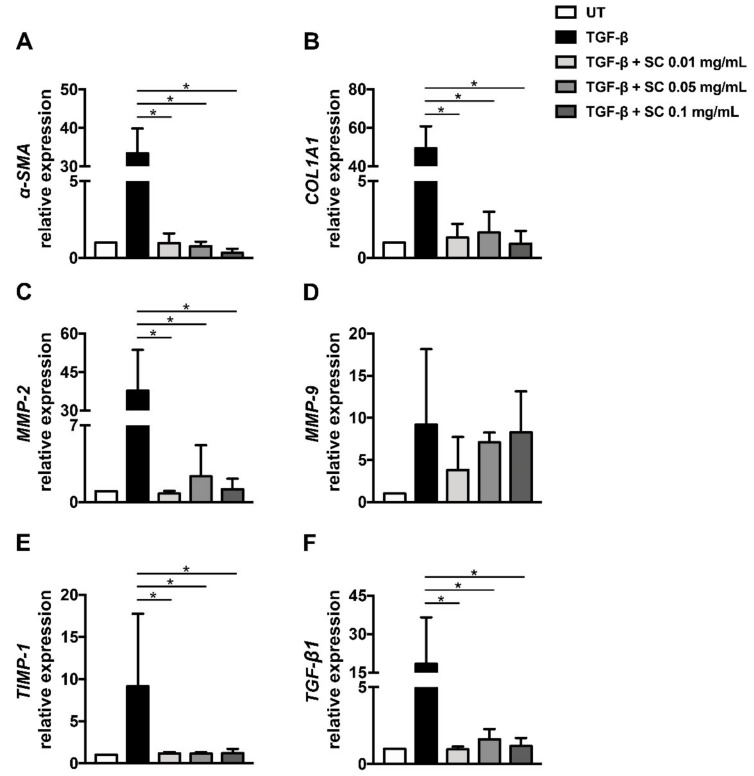
Effects of SC stem extract on mRNA expressions. The mRNA expression of (**A**) *α-SMA*, (**B**) *COL1A1*, (**C**) *MMP-2*, (**D**) *MMP-9*, (**E**) *TIMP-1*, and (**F**) *TGF-β1* was measured by quantitative RT-PCR. *GAPDH* was used as an internal control. UT: untreated; TGF-β: TGF-β1 2 ng/mL; SC: *Salacia chinensis* L. extract. * *p* < 0.05 indicates statistical significance from TGF-β1-treated group.

**Figure 4 ijms-20-06314-f004:**
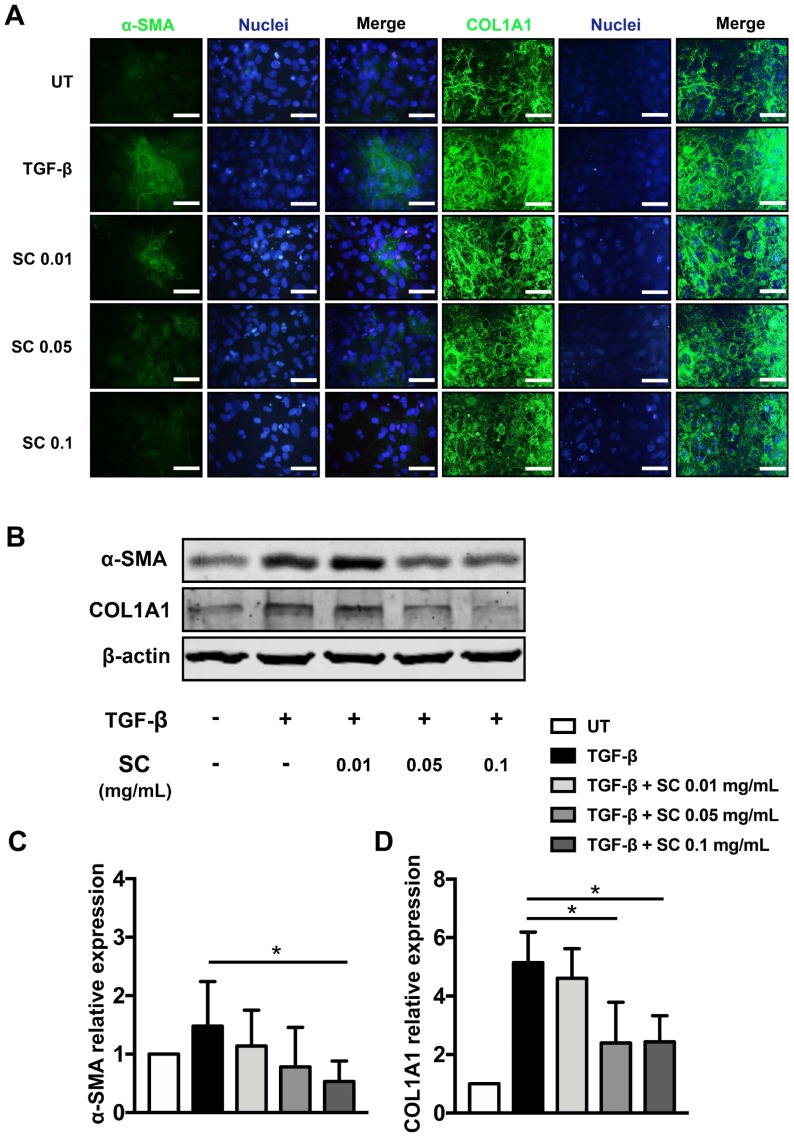
Effects of SC stem extract on α-SMA and collagen type I production. Both α-SMA and COL1A1 protein expression (Green) were examined by immunofluorescent staining (**A**). The nuclei (Blue) were stained with Hoechst 33342. The quantitative protein production of α-SMA and collagen type I (COL1A1) (**B**) was measured by western blotting. The bar graphs represent the relative expression of these proteins after normalization to β-actin (**C**,**D**). UT: untreated; TGF-β: TGF-β1 2 ng/mL; SC: *Salacia chinensis* L. extract. * *p* < 0.05 indicates statistical significance from the TGF-β1-treated group. Scale bar = 200 um.

**Figure 5 ijms-20-06314-f005:**
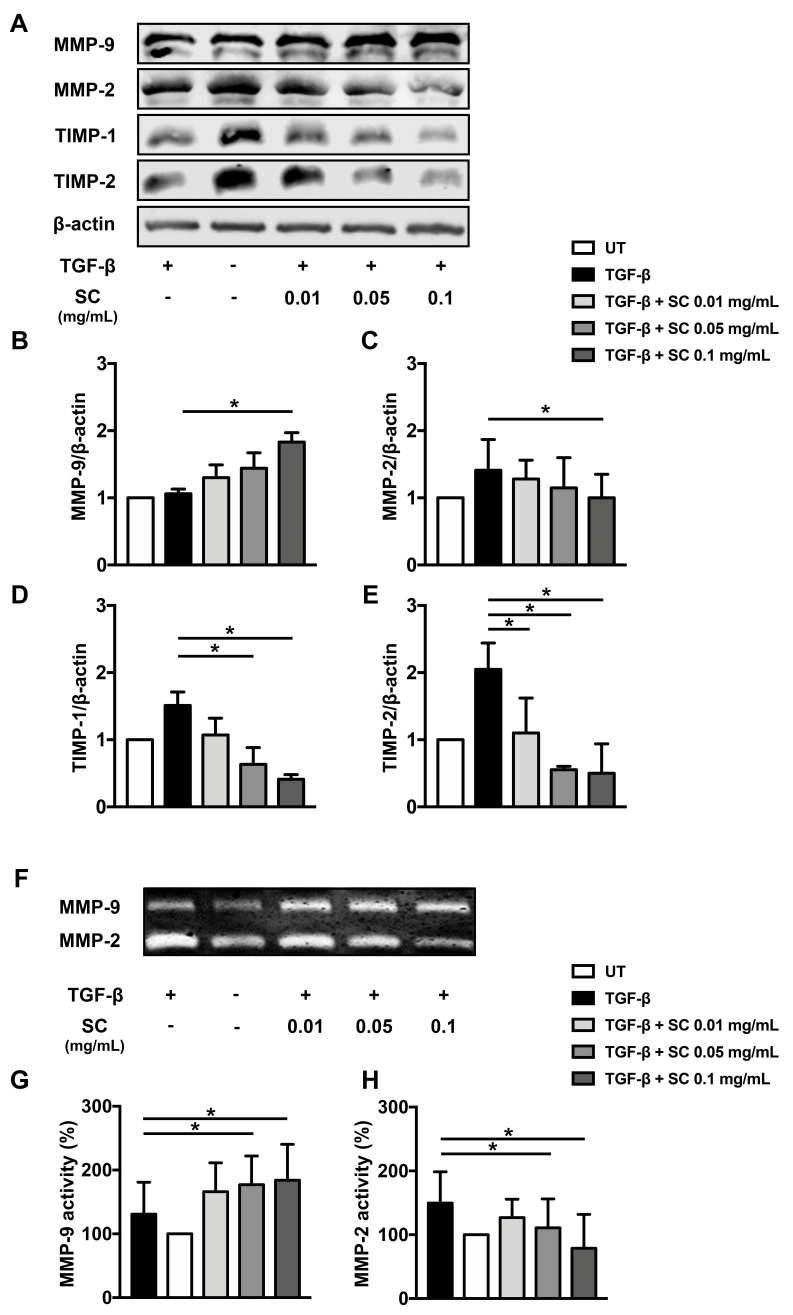
Effects of SC stem extract on secreted extracellular matrix components. Western blot analysis of MMP-9, MMP-2, TIMP-1, and TIMP-2 protein expressions in supernatant of LX-2 cells was shown (**A**). The bar graphs represent the relative protein expressions of MMP-9 (**B**), MMP-2 (**C**), TIMP-1 (**D**), and TIMP-2 (**E**) after normalization to β-actin. MMP-2 and MMP-9 activity was investigated by gelatin zymography (**F**) and the bar graphs represent the percentage of relative MMP-9 (**G**) and MMP-2 (**H**) activities. UT: untreated; TGF-β: TGF-β1 2 ng/mL; SC: *Salacia chinensis* L. extract. * *p* < 0.05 indicates statistical significance from TGF-β1-treated group.

**Figure 6 ijms-20-06314-f006:**
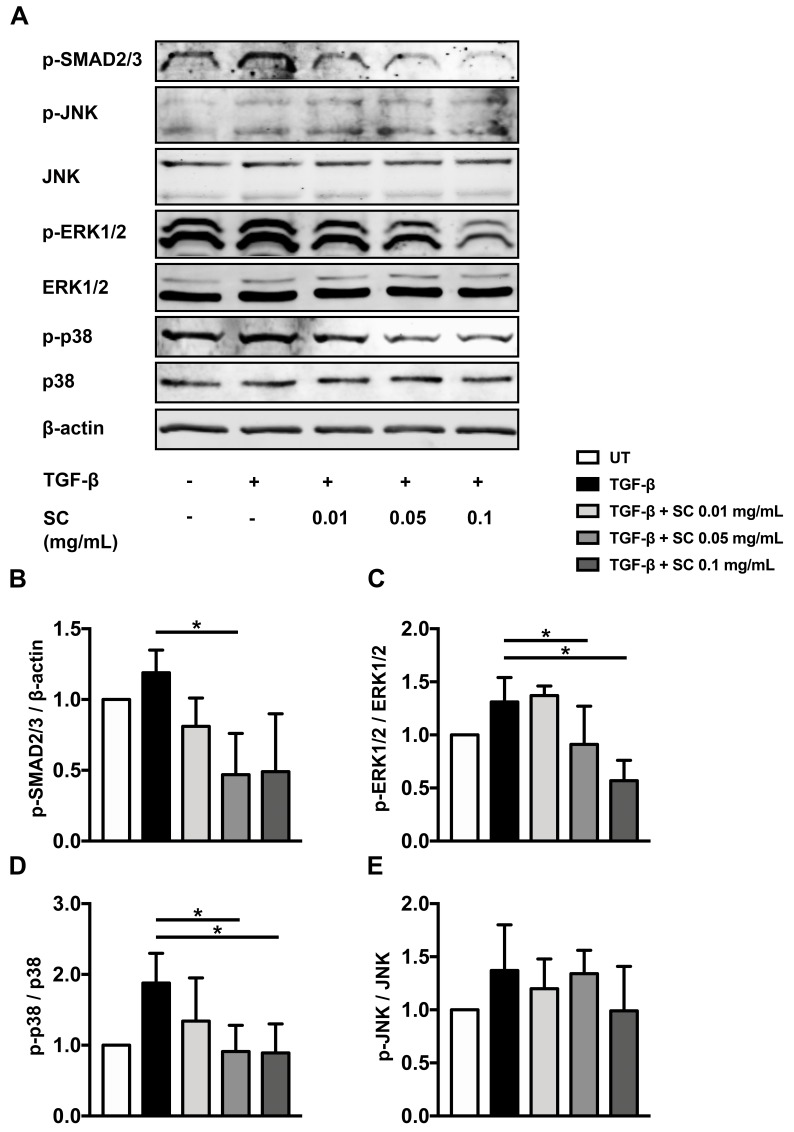
Effects of SC stem extract on phosphorylation of SMAD2/3, ERK1/2, p38, and JNK kinase. Western blot analysis of phosphorylated SMAD2/3, phosphorylated ERK1/2, ERK1/2, phosphorylated p38, p38, phosphorylated JNK, and JNK proteins are shown in (**A**). The bar graphs represent the relative expression of phosphorylated SMAD2/3 to β-actin (**B**), phosphorylated ERK1/2 to ERK1/2 (**C**), phosphorylated p38 to p38 (**D**), and phosphorylated JNK to JNK (**E**). UT: untreated; TGF-β: TGF-β1 2 ng/mL; SC: *Salacia chinensis* L. extract. * *p* < 0.05 indicates statistical significance from TGF-β1-treated group.

**Table 1 ijms-20-06314-t001:** Fold change of protein production (MMP-1, MMP-2, and MMP-9) and the ratio of production.

Groups	Fold Change of Protein Production					Ratio of Production		
	MMP-1	MMP-2	MMP-9	TIMP-1	TIMP-2	$\frac{MMP-1}{TIMP-1}$	$\frac{MMP-2}{TIMP-2}$	$\frac{MMP-9}{TIMP-1}$
UT	1.00 ± 0.00	1.00 ± 0.00	1.00 ± 0.00	1.00 ± 0.00	1.00 ± 0.00	1	1	1
TGF-β	1.20 ± 0.08	1.41 ± 0.46	0.96 ± 0.07	1.5 ± 0.20	2.05 ± 0.39	0.80	0.69	0.64
TGF-β + SC0.01	1.20 ± 0.01	1.28 ± 0.28	1.30 ± 0.19	1.07 ± 0.25	1.10 ± 0.52	0.63	0.62	1.27
TGF-β + SC0.05	1.17 ± 0.13	1.15 ± 0.45	1.44 ± 0.23	0.63 ± 0.25	0.55 ± 0.05	1.85 *	2.08	2.29
TGF-β + SC0.1	0.77 ± 0.07	1.00 ± 0.35	1.83 ± 0.14	0.41 ± 0.07	0.50 ± 0.44	1.87 *	2.00	4.46 *

* *p* < 0.05 indicates a statistically significance difference from the TGF-β1-treated group. UT: untreated; TGF-β: TGF-β1 2 ng/mL; SC: *Salacia chinensis* L. stem extract.

**Table 2 ijms-20-06314-t002:** Primers for qRT-PCR.

Genes	Forward Primer (F)	Reverse Primer (R)
*α-SMA*	5′-CGCATCCTCATCCTCCCT-3′	5′- GGCCGTGATCTCCTTCTG-3′
*COL1A1*	5′-GTCGAGGGCCAAGACGAAG-3′	5′-CAGATCACGTCATCGCACAAC-3′
*MMP-2*	5′-ACATCAAGGGCATTCAGGAG-3′	5′-GCCTCCGTATACCGCATCAAT-3′
*MMP-9*	5′-CCCGGAGTGAGTTGAACCA-3′	5′-GGATTTACATGGCACTGCCA-3′
*TIMP-1*	5′-CTTCTGCAATTCCGACCTCGT-3′	5′-CCCTAAGGCTTGGAACCCTTT-3′
*TGF-β1*	5′-GGCCAGATCCTGTCCAAGC-3′	5′-GTGGGTTTCCACCATTAGCAC-3′
*GAPDH*	5′-ATGGGGAAGGTGAAGGTCG-3′	5′-GGGGTCATTGATGGCAACAATA-3′

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
