# Peer review of "Salacia chinensis L. Stem Extract Exerts Antifibrotic Effects on Human Hepatic Stellate Cells through the Inhibition of the TGF-β1-Induced SMAD2/3 Signaling Pathway"

_ijms, 2019, doi:10.3390/ijms20246314_

Round 1

Reviewer 1 Report

The study of Phaosri et al. describes the anti-fibrotic effect of Salacia chinensis exerts on LX-2 cells. Anti-fibrotic therapy is an urgent clinical need. Therefore, the search for new anti-fibrotic agents is relevant. On the other hand, the suggested effective compound of the Salacia chinensis exerts, gallic acid, has been shown to be anti-fibrotic.

I have the following comments:

Salacia chinensis exerts have only been tested in LX-2 cells, not on other cells. No in vivo effect of the exerts is presented. Can the gallic acid or the Salacia chinensis exerts be administered orally? In the experiments the effects of Salacia chinensis exerts should be examined in the absence of TGFβ. In some experiments Salacia chinensis exerts inhibit below the untreated control.

In each panel, the number of independent experiments needs to be shown.

Author Response

Review 1

Salacia chinensis exerts have only been tested in LX-2 cells, not on other cells. No in vivo effect of the exerts is presented. Can the gallic acid or the Salacia chinensis exerts be administered orally? In the experiments the effects of Salacia chinensis exerts should be examined in the absence of TGFβ. In some experiments Salacia chinensis exerts inhibit below the untreated control.

Response1

Thank you reviewer for the comment. During fibrogenesis, the hepatic stellate cells (HSCs) are activated and produce tremendous amount of alpha-SMA and collagen. Since the molecular mechanism of hepatic fibrogenesis is strongly related to HSCs [1,2], we therefore focused on the effect of Salacia chinensis in HSCs, LX-2 cells, instead of hepatocytes. However, there is a study of methanolic extract of SC leaves showing the hepatoprotective effects [3]. Up to date, the stem of SC has been used as an ingredient in the folk Thai remedies, particularly at Prapokklao Hospital, Chanthaburi province, Thailand for liver cirrhosis treatment by oral administration. Even though the SC stem water extract has been used in liver cirrhosis patients, there is no scientific evidence supporting the molecular mechanism of its anti-fibrotic effect. We therefore started the investigation with a HSC cell line, and it is very interesting to further expand the experiments in the in vivo levels.

The reason we chose to conduct all experiments in the presence of TGF-β1 is that 1) the anti-fibrotic effect is clearly seen after TGF-β1 induction 2) TGF-β1 is considered as the main cytokine present in liver fibrosis.

As the reviewer recommended, we have conducted some experiments (Effect of SC on alpha-SMA production and zymogram of MMP-2 and MMP-9) in the absence of TGF-β1 and added the data in the supplemental data (Figure S2). As expected, the SC extract slightly reduced alpha-SMA production but did not clearly alter the MMP-2 and MMP-9 activities in the absence of TGF-β1.

Figure S2 shown in attachment.

In each panel, the number of independent experiments needs to be shown.

Response 2 The number of each independent experiment is shown in figure legends.

References

1 Xu, J.; Liu, X.; Koyama, Y.; Wang, P.; Lan, T.; Kim, I.G.; Kim H., I.H.; Ma, H.Y.; Kisseleva, T. The types of hepatic myofibroblasts contributing to liver fibrosis of different etiologies. Front. Pharmacol. 2014, 5 JUL, 1–12.

2. Xu, R.; Zhang, Z.; Wang, F.S. Liver fibrosis: Mechanisms of immune-mediated liver injury. Cell. Mol. Immunol. 2012, 9, 296–301.

3. Nakamura, S.; Zhang, Y.; Matsuda, H.; Ninomiya, K.; Muraoka, O.; Yoshikawa, M. Chemical structures and hepatoprotective effects of constituents from the leaves of Salacia chinensis. Chem. Pharm. Bull. (Tokyo). 2011, 59, 1020–8.

Reviewer 2 Report

This is a potentially very interesting study showing the impact of the extract of a traditional medicinal plant on the in vitro pro-fibrotic activity of a human stellate cell line. The authors convincingly show that extracts of the Salacia chinensis L. dose-dependently inhibited the TGFβ-induced trans-differentiation of the cells to myofibroblasts as well as the TGFβ-dependent induction of a number of pro-fibrotic genes. Still, the study suffers from significant shortcomings:

Major
(1) Important controls are missing: What is the impact of the extract on all parameters determined in absence of TGFβ. These data need to be included for all parameters at all concentrations of the extract tested. At least in the case of MMP9 the data presented suggest that the extract might have an effect on its own.
(2) Concentration range and specificity: The authors used a concentration of up to 0.1 mg/ml of extract. Assuming a molecular weight of the active component(s) around 200, such a concentration would correspond to a total concentration of 500 µM of potentially active compounds. This is a fairly high concentration. Therefore, an adequate control must be included. Such a control could be an equally concentrated extract of a reference plant for which no anti-fibrotic effect is known.
(3) Along the same lines, specificity for the regulation of TGFβ target genes must be shown. The authors need to prove that the extract at the concentration chosen does not affect other cellular signaling chains, for example insulin signaling or signaling of other cytokines.

Finally, the description of the results concerning the MMP9 and MMP2 protein expression and activity in section partially are at odds with the results shown in the figures (lines 214 an following). Similarly, the labelling of the Fig. 5a seems to be wrong (TGFβ and control switched?) as it does not reflect the analysis shown in the subsequent column diagrams.

Minor
The HPLC analytics of the extract does not comply with current standards to identify specific components. The authors might consider including a more meaningful analysis of the components of their extract. Actually, an analysis of the chemical composition of different extracts of this plant has been published (PMID: 28955795 - cited in the manuscript in different context - and possibly PMID: 24730982).

Round 2

Reviewer 1 Report

The study of Phaosri is slightly improved. What still remains is that Salacia chinensis exerts have only been tested in LX-2 cell line, not in primary stellate cells, and no in vivo experiments are presented.

Different from your statement, the number of experiments are still not shown in each panel. It is only given in Fig. S2. Adding this to all figures is required.

Author Response

Reviewer#1

1) The study of Phaosri is slightly improved. What still remains is that Salacia chinensis exerts have only been tested in LX-2 cell line, not in primary stellate cells, and no in vivo experiments are presented.

We really thank the reviewer for your comment. We agree with the reviewer that extending the study in primary stellate cells and in animal model is crucial to make it more scientifically significant. However, our initial scope is to identify whether the extract has reasonable degree of potency as an anti-fibrotic agent for further carrying on the study in the more complex conditions including animal mode. We currently have a study plan to fulfill important aspects to ensure that this plant authentically possess anti-fibrosis and can be used at least in mammalian animals before stepping up to studies in humans.  

2) Different from your statement, the number of experiments are still not shown in each panel. It is only given in Fig. S2. Adding this to all figures is required.

We thank for the reviewer very much and apologize for the mistake. The number of each independent experiment has been added. The modifications have been shown below,

Page 8, line 136-140 : Figure 1: Effects of SC stem extract on stimulated HSCs morphology. Cell viability of LX-2 after SC treatments at 0.01 to 0.1 mg/mL concentration (n=6) was shown in (A). Schematic diagram of experimental design was shown in (B). LX-2 morphology observed under a phase contrast microscopy (n=6) was shown in (C). UT: untreated; TGF-β: TGF-β1 2 ng/mL; SC: Salacia chinensis L.extract. Scale bar = 50 µm.

Page 8 line 143-144 to page 9 line 145-146 : Figure 2: Effects of SC stem extract on HSCs 2D migration. Cell migration was observed at 0h, 24h and 48h after SC treatments (A). The percentage of cell migration at 24h and 48h after SC treatments (n=6) was quantified in (B). UT: untreated; SC: Salacia chinensis L.extract. Scale bar = 50 µm.”

Page 10 line 164-168 : “Figure 3: Effects of SC stem extract on mRNA expressions. The mRNA expressions of (A) α-SMA, (B) COL1A1, (C) MMP-2, (D) MMP-9, (E) TIMP-1, and (F) TGF-β1 was measured by quantitative RT-PCR (n=3). GAPDH was used as an internal control. UT: untreated; TGF-β: TGF-β1 2 ng/mL; SC: Salacia chinensis L.extract. *p < 0.05 indicates statistical significance from TGF-β1-treated group.

Page 11 line 188-192 : Figure S2 Western blot analysis of α-SMA protein expressions in LX-2 cells (n=6) was shown (A). The bar graphs represent the relative protein expressions of α-SMA (B), after normalization to β-actin. MMP-2 and MMP-9 activity was investigated by gelatin zymography (n=6) (C) and the bar graphs represent the percentage of relative MMP-9 (D) and MMP-2 (E). SC: Salacia chinensis L.extract.

Page 12 line 196-203 : “Figure 4: Effects of SC stem extract on α-SMA and collagen type I production. Both α-SMA and COL1A1 protein expression (Green) were examined by immunofluorescent staining (n=3) (A). The nuclei (Blue) were stained with Hoechst 33342. The quantitative protein productin of α-SMA and collagen type I (COL1A1) (B) were measured by western bloting (n=3). The bar graphs represent the relative expression of these proteins after normalization to β -actin (C-D). UT: untreated; TGF-β: TGF-β1 2 ng/mL; SC: Salacia chinensis L.extract. *p < 0.05 indicates statistical significance from TGF-β1-treated group. Scale bar = 200 um.”

Page 15 line 245-247 to page 16 line 248-252 : “Figure 5: Effects of SC stem extract on secreted extracellular matrix components. Western blot analysis of MMP-9, MMP-2, TIMP-1, and TIMP-2 protein expressions in supernatant of LX-2 cells was shown (A). The bar graphs represent the relative protein expressions (n=3) of MMP-9 (B), MMP-2 (C), TIMP-1 (D) and TIMP-2 (E) after normalization to β-actin. MMP-2 and MMP-9 activity was investigated by gelatin zymography (F) and the bar graphs represent the percentage of relative MMP-9 (G) and MMP-2 (H) activities (n=3).UT: untreated; TGF-β: TGF-β1 2 ng/mL; SC: Salacia chinensis L.extract. *p < 0.05 indicates statistical significance from TGF-β1-treated group.

Page 17 line 265-272 : “Figure 6: Effects of SC stem extract on phosphorylations of SMAD2/3, ERK1/2, p38 and JNK kinase. Western blot analysis of phosphorylated SMAD2/3, phosphorylated ERK1/2, ERK1/2, phosphorylated p38, p38, phosphorylated JNK and JNK proteins was shown in (A). The bar graphs represent the relative expression of phosphorylated SMAD2/3 to β -actin (B), phosphorylated ERK1/2 to ERK1/2 (C), phosphorylated p38 to p38 (D) and phosphorylated JNK to JNK (n=3) (E). UT: untreated; TGF-β: TGF-β1 2 ng/mL; SC: Salacia chinensis L.extract. *p < 0.05 indicates statistical significance from TGF-β1-treated group.

Supplemental Figure : Figure S3 Western blot analysis of ?-SMA protein expressions in various kinds of plant extraction (A). The bar graphs represent the relative protein expressions of ?-SMA (n=3) (B), after normalization to β-actin. UT = untreated, TGF-β = TGF-β1 2 ng/mL, SS = Smilax sp, SG = Smilax glabra, SCO = Smilax corbularia, EE = Erycibe expansa, DS = Derris scandens Benth, SC = Salacia chinensis L. *p < 0.05 indicates statistical significance from TGF-β1-treated group.

Reviewer 2 Report

The response of the authors to the concerns is adequate. Additional experiments were performed that fill the gaps. Unfortunately, the new supplementary figure S3 is missing in the file with the supplementary data. Please append. 

Round 3

Reviewer 1 Report

The authors have sufficiently answered my comments.